# Related in Death? Further Insights on the Curious Case of Bishop Peder Winstrup and His Grandchild's Burial

**Maja Krzewińska** [1,2,*], **Ricardo Rodríguez-Varela** [1,2], **Reyhan Yaka** [1,2], **Mário Vicente** [1,2], **Göran Runfeldt** [3], **Michael Sager** [3], **Caroline Ahlström Arcini** [4], **Torbjörn Ahlström** [5], **Niklas Hertzman** [6], **Jan Storå** [7] and **Anders Götherström** [1,2,*]

1 Centre for Palaeogenetics, 106 91 Stockholm, Sweden
2 Department of Archaeology and Classical Studies, Stockholm University, 106 91 Stockholm, Sweden
3 FamilyTreeDNA, Gene by Gene, Houston, TX 77008, USA
4 The Archaeologists, National Historical Museums, 114 84 Stockholm, Sweden
5 Department of Archaeology and Ancient History, Lund University, 221 00 Lund, Sweden
6 Södra Strandgatan 19 1204, 252 24 Helsingborg, Sweden
7 Osteoarchaeological Research Laboratory, Stockholm University, 106 91 Stockholm, Sweden
* Correspondence: maja.krzewinska@arklab.su.se (M.K.); anders.gotherstrom@arklab.su.se (A.G.)

**Abstract:** In 2021, we published the results of genomic analyses carried out on the famous bishop of Lund, Peder Winstrup, and the mummified remains of a 5–6-month-old fetus discovered in the same burial. We concluded that the two individuals were second-degree relatives and explored the genealogy of Peder Winstrup to further understand the possible relation between them. Through this analysis, we found that the boy was most probably Winstrup's grandson and that the two were equally likely related either through Winstrup's son, Peder, or his daughter, Anna Maria von Böhnen. To further resolve the specific kinship relation, we generated more genomic data from both Winstrup and the boy and implemented more recently published analytical tools in detailed Y chromosome- and X chromosome-based kinship analyses to distinguish between the competing hypotheses regarding maternal and paternal relatedness. We found that the individuals' Y chromosome lineages belonged to different sub-lineages and that the X-chromosomal kinship coefficient calculated between the two individuals were elevated, suggesting a grandparent–grandchild relation through a female, i.e., Anna Maria von Böhnen. Finally, we also performed metagenomic analyses, which did not identify any pathogens that could be unambiguously associated with the fatalities.

**Keywords:** Winstrup; kinship; aDNA; Y chromosome; aMeta

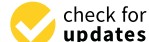



## 1. Introduction

Peder Pedersen Winstrup (1605–1679) was a prominent figure in the Protestant church and was an accomplished scholar. He lived in both Denmark and Sweden at various stages of his life, and, following his demise, he was interred in Lund Cathedral [1–3]. In the early 19th (and then 20th) century, his coffin was unsealed, and there have been accounts of remarkably well-preserved remains. (Figure 1A). In 2012, when Winstrup's coffin was being moved to a cemetery outside the church, an opportunity arose to initiate a multidisciplinary research project in collaboration with the church, the Historical Museum, and Lund University [3]. The project involved various analyses of Winstrup's remains and the contents of his coffin, including CT scanning and X-ray, genetic and epidemiological analyses, and examinations of his clothing, artifacts, and the plant and insect remains found in the coffin (e.g., [3–5]).

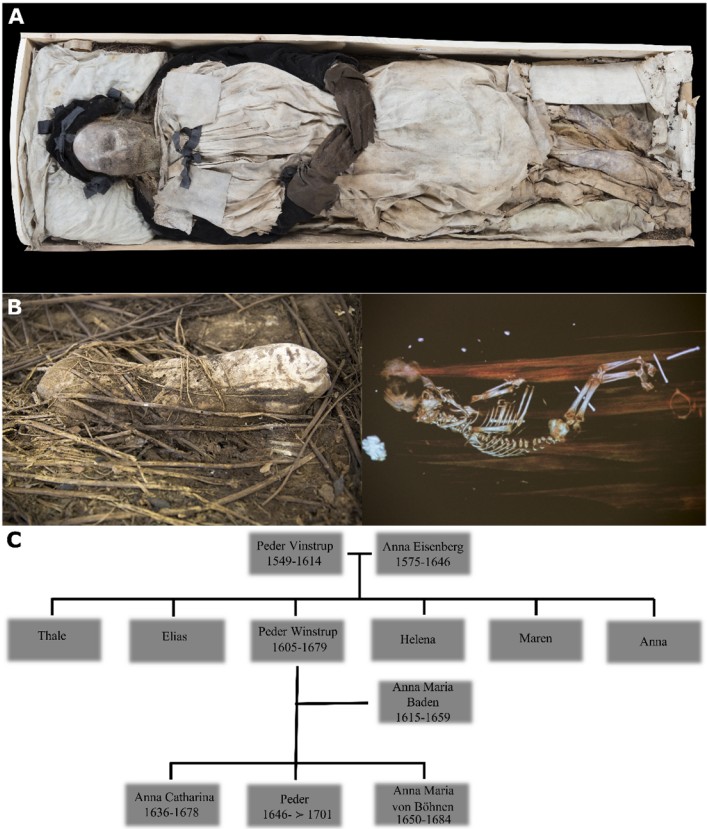

**Figure 1.** (**A**) Mummified body of bishop Peder Winstrup in his coffin; (**B**) the bundle containing mummified remains of the fetus and a CT scan image of the contents; (**C**) genealogical reconstruction of Peder Winstrup's family tree. Images adapted from [1].

In the course of extensive research, mummified remains of a 5–6-month-old human fetus placed below Winstrup's right tibia were discovered [3] (Figure 1B). The surprising find sparked a discussion concerning the possible kin relationship between Winstrup and the fetus. As it was fairly common for children to be buried with unrelated adult individuals during the Middle Ages [6,7] it was deemed highly probable the two were unrelated. This appeared especially plausible as the fetus was seemingly hurriedly placed in the coffin, hidden underneath the silken lining and displacing the bishop's legs in the process. However, seeing that the coffin was placed in a vaulted family tomb, it could not be ruled out that the fetus was placed in the coffin sometime later, possibly by a relative of Peder Winstrup.

We have explored the question using genomic data and concluded the two were second-degree relatives [1]. Detailed genealogical analyses of the bishop Peder Winstrup's family tree (Figure 1C) revealed the most feasible 2nd-degree-kin relation in this case would be that of grandparent-grandchild. As Peder himself had both daughters and a son, we were unable to distinguish conclusively between the maternal and paternal relation using the tools implemented at the time. To further enhance the analyses, we generated more sequencing data from the best performing libraries and used a novel approach to Y chromosome identification. We also estimated kinship between two individuals using two recently developed tools, *KIN* [8] and *NgsRelate* [9], where the latter allowed us to focus on X chromosomal degree of relatedness (*NgsRelate*). Furthermore, we performed metagenomic screening with *aMeta* [10]. This second analysis provided new insights into the kinship relation of the individuals.

## 2. Materials and Methods

### 2.1. Wet Laboratory Processing

The sample handling and processing procedures, outlined in detail in [1], were conducted as part of the earlier published study. In brief, the steps involved anthropological analysis utilizing a CT scan generated at Lund University Hospital [5]. The DNA was extracted from the right femur of the Bishop and the left femur of the fetus. Both bones had traces of desiccated soft tissue (periosteum), which was also used in a separate DNA extraction. The procedure was undertaken at the clean ancient DNA facilities at the Archaeological Research Laboratory, Stockholm University. Samples were decontaminated via UV irradiation (254 nm) at approximately J/cm$^2$ per side. The hard tissues (bones) were drilled with a Dremel tool at 5000 rpm, while fragments of soft tissue were removed with decontaminated forceps. Thereafter, the DNA was extracted from 93 mg (Winstrup) and 46 mg (fetus) of bone powders and from 7 mg (Winstrup) and 8 mg (fetus) of collected fragments of soft tissue covering the skeletal elements sampled.

We used our standard DNA extraction and purification protocols [11,12], followed by preparation of double-stranded DNA libraries [13] and qPCR validation. The libraries were indexed in a $3 \times 50$ µL PCR reaction. Finally, libraries were purified with magnetic beads (AmPure XP, Beckman Coulter, Indianapolis, IN, USA) and quantified on a bioanalyzer (Agilent Technologies$^{TM}$, Sundbyberg, Sweden). The libraries were shotgun-sequenced both on the Illumina HiSeq X and the NovaSeq 6000 (Illumina, Inc., San Diego, CA, USA). All sequencing was performed at the SciLifeLab DNA NGI sequencing facility, Stockholm, Sweden.

### 2.2. In Silico Processing

After de-multiplexing at NGI (https://github.com/NationalGenomicsInfrastructure/ngi_pipeline (accessed on 6 June 2021)), the forward and reverse pair-end fastq reads were merged with 11 bp overlap and trimmed with cutadapt v. 2.3, Adapter Removal v. 2.1.7 [14], or MergeReadsFastq_cc.py script [15]. The BWA aln (-l 16500 -n 0.01 -o 2) v. 0.7.13 [16] was used to map the sequences to the human reference genome build 37 (version hs37d5), and the FilterUniqSAMCons_cc.py [15] was used for PCR duplicate removal. A minimum matching of 90% to the reference genome, a minimum read length of 35 bp, and a minimum mapping quality of 30 were requirements for reads to be incorporated in further analyses. The individual sex was estimated using *Ry* ratio [17].

We verified the presence of aDNA templates [18–22] using PMDtools [23]. Finally, the levels of contamination were measured both in the mitochondrial DNA [24,25] and in the X chromosome (Angsd; [26]). Mitochondrial DNA (mtDNA) haplogroups were called using HaploGrep 2.1.16 [27] on consensus mtDNA genomes generated with mpileup and samtools software (version 1.5) [28], including only sequences of a minimum mapping score and base quality of 30.

### 2.3. Y Chromosome Analyses

For Y chromosome lineage identification, we adopted a two-fold approach: we used the newly published pathPhynder method [29] and the methodology utilized by the R&D Team at FamilyTreeDNA (FTDNA) specializing in Y-chromosomal analyses and Y tree refinement. The latter approach involved putting less weight on SNPs in certain parts of the Y chromosome and also to G>A and C>T mutations in aDNA, while considering all published and unpublished SNP variants available in the FTDNA database. The methodology was earlier described in [30]. In short, Y chromosome reads were aligned to hg38 using BWA-ALN, while mapDamage2.0 was used to downscale the base quality of transitions in aDNA reads. Thereafter, base and map qualities of 30 were used in variant calling. Haplogroup assignment was completed by identification of known branch-defining variants while putting less weight on non-private mutations identified as highly recurrent variants in either modern and ancient datasets and on variants occurring in problematic Y chromosome regions (e.g., the centromere, DYZ19 repeat, and Yq12 heterochromatic region).

Secondly, we used pathPhynder (2020-12-19-b8532c0) best path method with BigTreeY SNP data provided with the program (BigTree.Y.201219.vcf.gz) [29]. We first trimmed the sequencing reads 10 bp at both ends and used both transitions and transversions in the "best path method" with default settings, i.e., base quality of 20, mapping quality of 25, and removing singleton transversions. We then used untrimmed BAM files, restricting the analyses to only transversion SNPs. We ran the analyses twice, setting the maximum number of alternative alleles tolerated while traversing the tree to the default 3 and to a very broad 100 (Table S3).

*2.4. Kinship Analysis*

In order to estimate kinship between the studied individuals, first we used NgsRelate software (version 2.0) [9]. We ran NgsRelate using BAM files as input with default parameters using a panel of 1,554,712 autosomal transversion SNPs from the Estonian Genome Diversity Project (EGDP) [31]. We calculated population allele frequencies from two different datasets of individuals from various archeological contexts in Sweden: Viking Age to Medieval dataset (*n* = 163) dataset covering three different sites, i.e., Sigtuna (*n* = 64), Fjälkinge (*n* = 11), and Västerhus (*n* = 88) [7,32], and a 17th century dataset (*n* = 44) [33]. All samples in the ancient reference panels were generated in a similar manner as the individuals tested here. We also tested different SNP filtering with various settings, including with minimum allele frequencies of 0.15 (15%), 0.10 (10%), and 0.05 (5%), using the above-mentioned ancient DNA reference panels. Secondly, we used KIN [8] to further verify the degree of relatedness between the tested individuals (Table S4).

Finally, to infer pedigree relationship between 2nd-degree related individuals, we ran *NgsRelate* on X-chromosomal loci using a panel of 74,045 transversion SNPs from EGDP, following the same method described above.

*2.5. ROH*

Elevated consanguinity levels can influence kinship coefficients (k0, k1, k2) [34], therefore it is important to take this information into account before inferring kinship relations.

We employed hapROH (v 3.0), a software developed by Ringbauer et al. in 2021, to compute regions of homozygosity (ROH) in ancient DNA data. We executed hapROH following the guidelines provided by the authors at https://pypi.org/project/hapROH/ (accessed on 15 November 2023) [35].

*2.6. Metagenomic Analyses*

The metagenomic screening was performed using an aMeta v.1.0.0 pipeline [10]. We restricted the analyses to the first step, which involved using a KrakenUniq *k*-mer classifier [36]. This allowed to screen for potential pathogenic microbes using a custom query database containing all NCBI NT microbial sequences (bacteria, fungi, and viruses) and selected eukaryotic genomes [10]. KrakenUniq outputs were then filtered to exclude microbial species with less than 1000 *k*-mers and 200 taxReads to minimize false positives. Based on the outputs, we calculated *k*-mer to taxReads proportion and used a conservative cut-off of 9 and average read length below 120 bp as means to identify authentic ancient microorganisms (Table S6).

The computations were performed on resources provided by NAISS/SNIC from the Uppsala Multidisciplinary Center for Advanced Computational Science (UPPMAX) [37].

**3. Results**

As previously described, we generated and merged data from bone and soft tissue. For resequencing, we used libraries generated from bone samples as they had higher indigenous content then the soft tissue [1]. We sequenced the two libraries in 1/13 of a NovaSeq S4 lane, resulting in an additional 2,890,673 reads in win001 and 12,147,341 reads in win002. In accordance with published protocols, we used the consistent mtDNA results obtained across different extractions and library builds as proof of sample integrity and

then merged the total of nine files for each of the two individuals tested, resulting in the final genome coverage of 0.78× for the adult and 0.98× for the fetus. Mitochondrial DNA (mtDNA) coverage varied significantly and was 9487.38× in the adult and 812.27× in the fetal remains. MtDNA haplogroups were assigned using HaploGrep 2 (v2.4.0) [27], confirming that the two males were carriers of different mitochondrial DNA lineages (Tables 1 and S3).

**Table 1.** Summary statistics.

| Sample_ID | Genome Coverage | MtDNA Coverage | Biol. Sex | MtDNA Hg | FTDNA Y Hg | pathPhynder Y Hg |
|---|---|---|---|---|---|---|
| Winstrup | 0.79 | 9487.38 | XY | H3b7 | R-Z209 (R-BY54766) | R1b1a1b1a1a2a1a1a1~ |
| Fetus | 0.96 | 812.27 | XY | U5a1a1 + 152 | R-DF17 (R-BY1806) | R1b1a1b1a1a2a1a2~ |

### 3.1. Y Chromosome Analyses

The Y chromosome was assigned using the FTDNA algorithm and the best path pathPhynder method (Tables 1 and S3) [29]. As a result of additional sequencing, we generated 25,398 more Y chromosome reads (a 7.5% increase) for individual win001 and 100,301 more reads (36.9% increase) for individual win002. The first approach, relying on detailed analyses of presence and absence of branch-supporting SNPs, showed that, other than carrying SNPs supporting different branches, win001 was negative for four of the R-DF17 SNPs, while win002 was negative for four of the R-Z209 SNPs (Table S3).

The pathPhynder (2020-12-19-b8532c0) best path method results suggested that win001 belonged to lineage R1b1a1b1a1a2 or R1b1a1b1a1a2a1a1a1~, while win002 belonged to R1b1a1b1a1a2a1a2~, pointing to the possibility of the lineages being different (Tables 1 and S3).

### 3.2. Kinship Analyses

In order to better understand previously detected kinship, we employed additional previously not used tools, KIN [8] and *NgsRelate* [9], to estimate the degree of relatedness and possible pedigree relationship between the two individuals. For the *NgsRelate* allele frequency estimation, we used various ancient genomic reference datasets, all consisting of genomic data generated from ancient samples from Sweden dated to the medieval period (*n* = 163) and the 17th century (*n* = 44) [7,32,33]. Finally, we assumed a minimum of 5000 overlapping SNPs as a cut-off value, as this has been shown to be a reliable and conservative value allowing for kinship estimation up to the third-degree [38]. Based on obtained autosomal kinship coefficients ($\theta$) (0.09–0.15), associated $k0$ (0.31–0.58), $k1$ (0.36–0.56), and $k2$ (0.08–0.01) values (Tables 2 and S4), and results from KIN (Table S5), we noted that the two individuals were most probably second-degree relatives, as shown earlier [1]. The X-chromosomal kinship coefficient ($\theta$) calculated with *NgsRelate* resulted in elevated values (0.40–0.48), consistent with relatedness through a female (Table S4).

**Table 2.** Summary results of *NgsRelate* analyses testing the kinship relation between Peder Winstrup and the fetus with two different reference populations from Sweden: medieval (*n* = 163) and 17th century (*n* = 44) [7,33].

| Reference Panel | | NgsRelate (Autosomal Chromosomes) | | | | | | NgsRelate (X Chromosome) | |
|---|---|---|---|---|---|---|---|---|---|
| *N* | Maf | # SNPs | $k0$ | $k1$ | $k2$ | $\theta$ | | # SNPs | $\theta$ |
| 163 | 0.05 | 485,784 | 0.35663 | 0.521772 | 0.0178331 | 0.150689 | | 6293 | 0.490875 |
| 44 | 5 | 437,721 | 0.608852 | 0.364639 | 0.010957 | 0.09666 | | 4529 | 0.406657 |

*3.3. ROH*

We executed hapROH following the authors at https://pypi.org/project/hapROH/ (accessed on 15 November 2023) [35]. Our findings revealed that neither of the two analyzed individuals exhibited cumulative ROH lengths exceeding 3 cM.

*3.4. Metagenomic Analyses*

We performed metagenomic screening using an *aMeta* pipeline [10]. First, we generated KrakenUniq outputs [36] for individual libraries (Table S6). Numerous potentially pathogenic species were identified in the initial screening step. We then used a stringent filtering step to select the most probable ancient pathogens (*k*-mer to taxReads proportion $\geq 9$) before continuing with MALT [39]. As no unusual human pathogens were identified in any of the sequenced libraries after implementing the ratio threshold, the MALT database and mapping steps were omitted.

## 4. Discussion

Our previous findings suggested a second-degree relatedness (i.e., half-siblings, uncle–nephew, grandfather–grandson) between the fetus and bishop Peder Winstrup. As the two did not share maternally inherited mitochondrial DNA (H3b7 vs. U5a1a1 + 152, Tables 1 and S3A) the kin relation could only be paternal if we considered Winstrup's family history (Figure 1C). Peder had four sisters, all of whom married and settled in Zealand, and a brother [40], who died unmarried in 1633. We therefore excluded an uncle–nephew relation, as the infant could not have been any of Winstrup sisters' child, otherwise they would share mtDNA, while his brother, Elias, likely fathered no children at all. Furthermore, since both Winstrup's parents died decades before him, we considered the possibility of a half-sibling relation as highly unlikely. The most parsimonious explanation is that the fetus was Winstrup's grandson. In this case, the likely connection would be either through daughter Anna Maria von Böhnen or son Peder, both of whom were married and lived in Lund. As members of the bishop's immediate family, they would also likely have access to the family crypt.

To test this hypothesis, we first reanalyzed the Y chromosome haplotypes and showed that win001 belonged to lineage R-Z209 (R-BY54766), while win002 was confirmed to belong to R-DF17 (R-BY1806). Both are different lineages of R-Z274, which are estimated to be about 4100 years old. The findings were further confirmed by results from pathPhynder, suggesting the two lineages were closely related but not identical, i.e., R1b1a1b1a1a2a1a1a1~ (win001) and R1b1a1b1a1a2a1a2~ (win002) (Tables 1 and S3B,C). This finding would align better with the explanation that win002 was more likely one of Peder Winstrup daughters' children. Therefore, we next used *NgsRelate* and *KIN* software to estimate autosomal $\theta$, and *NgsRelate* to estimate X-chromosomal $\theta$ between the two individuals. We inferred this pair as grandfather–grandson due to their relatively high X-chromosomal $\theta$ (Tables 1 and S4). We also noted that X-chromosomal $\theta$ (0.40) between the two individuals was higher than expected; however, this higher value of the X-chromosomal $\theta$ could be explained by randomness in recombination, as the value of 0.25 represented the average expectation [34]. Similarly, inflated X-chromosomal $\theta$ could potentially reflect the overall smaller number of SNPs from the X chromosome. However, this value would be null if the two were related through Peder junior. An alternative explanation of the observed elevated X-chromosomal $\theta$ could involve inbreeding; however, hapROH analysis revealed there were no notable levels of inbreeding in the parents of the bishop and the fetus.

Taken together, the evidence suggests it is likely that the child was that of Anna Maria von Böhnen. Anna Maria died in 1684, only five years after Winstrup, and, in 1837, it was discovered by Nils Henrik Lovén that she likely died during childbirth [3]. If the boy was indeed Anna Maria's, his remains may have been placed in the coffin sometime after Winstrup's death.

Finally, the metagenomic screening did not detect any unusual pathogenic species. Although the femur is not an optimal skeletal element for mining for ancient bacteria, we performed the aMeta screening to obtain a general microbiological profile of the tested samples. We identified numerous environmental microorganisms and plant pathogens, which could probably be associated with the decaying plant material deposited in the coffin. We also found a number of potentially pathogenic strains known to cause infections in humans, many of which are a usual part of human microbiota (oral, gut, and skin flora) or are widespread in the environment, i.e., *Histoplasma capsulatum*, *Nocardia nova*, *Nocardia otitidiscaviarum*, *Staphylococcus aureus*, *Staphylococcus epidermidis*, *Acinetobacter johnsonii*, *Enterococcus faecalis*, *Streptococcus sanguinis*, *Acinetobacter lwoffii*, and numerous species of *Pseudomonas*, etc. (Table S6). Perhaps unsurprisingly, we found no evidence of *Mycobacterium tuberculosis*, despite the fact Sabin et al. (2020) [41] used samples taken from Peder Winstrup to generate the highest coverage ancient genome of *Mycobacterium tuberculosis* to date. This false negative emphasizes the difficulties associated with ancient metagenomic analyses and the need for appropriate sampling strategies when attempting to investigate specific diseases. Here, samples taken from the femur vs. the lung nodule resulted in significantly different microbiological profiles.

**Supplementary Materials:** The following supporting information can be downloaded at: https://www.mdpi.com/article/10.3390/heritage7020027/s1, Table S1. Updated sequence statistics; Table S2. Contamination estimates of merged genomic libraries from individuals presented in this study; Table S3. Uniparental haplogroup estimates; Table S4. NgsRelate results summary; Table S5. KIN analysis results where the two tested individuals were compared to the 17th century reference panel (N = 44); Table S6. KrakenUniq results and metrics obtained from the different genomic libraries.

**Author Contributions:** Conceptualization, M.K., J.S., A.G., T.A. and C.A.A.; methodology, G.R., M.S., R.R.-V., R.Y. and M.V.; software, R.R.-V., R.Y. and M.V.; validation, R.R.-V., R.Y., G.R., M.S. and M.K.; formal analysis, M.K., R.R.-V., R.Y., G.R., M.S. and M.V.; investigation, T.A. and N.H.; resources, A.G.; data curation, M.V.; writing—original draft preparation, M.K.; writing—review and editing, M.K., with input from all authors; visualization, M.K., C.A.A., T.A. and N.H.; supervision, A.G. and J.S.; funding acquisition, A.G. and M.K. All authors have read and agreed to the published version of the manuscript.

**Funding:** This project was partially supported by Riksbankens Jubileumsfond (grant no. P16-0553:1) Erik Philip-Sörensen Foundation, the Crafoord Foundation, the Knut and Alice Wallenberg Foundation (1000 Ancient Genomes Project Grant 2016), and the Swedish Research Council (2013-4959, 2019-00849). The computations were enabled by resources provided by the National Academic Infrastructure for Supercomputing in Sweden (NAISS) and the Swedish National Infrastructure for Computing (SNIC) at UPPMAX partially funded by the Swedish Research Council through grant agreements no. 2022-06725 and no. 2018-05973. The analyses were performed under the following projects: SNIC 2021/22-182, and naiss2023-22-378.

**Data Availability Statement:** The genome data can be downloaded from the European Nucleotide Archive (ENA) using the following project accession numbers PRJEB43107 and PRJEB71912.

**Acknowledgments:** We thank Per Karsten for initiating this project, Gunnar Menander for photographs, and Carolina Bernhardsson from the Department of Organismal Biology at Uppsala University for help with the processing and curation of data. The authors acknowledge support from the National Genomics Infrastructure in Stockholm funded by Science for Life Laboratory, the Knut and Alice Wallenberg Foundation and the Swedish Research Council, and SNIC/NAISS/Uppsala Multidisciplinary Center for Advanced Computational Science for assistance with massively parallel sequencing and access to the UPPMAX computational infrastructure.

**Conflicts of Interest:** Author Göran Runfeldt and Michael Sager were employed by FamilyTreeDNA, Gene by Gene. The remaining authors declare that the research was conducted in the absence of any commercial or financial relationships that could be construed as a potential conflict of interest.

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
