# Peer review of "Related in Death? Further Insights on the Curious Case of Bishop Peder Winstrup and His Grandchild’s Burial"

_heritage, doi:10.3390/heritage7020027_

Round 1

Reviewer 1 Report

Comments and Suggestions for Authors

KrzewiÅ„ska and coauthors presented results of additional sequencing effort on previously generated libraries from the remains of Peder Winstrup and the foetus found in the same coffin. 

Genetic analysis of these individuals was published previously, but the newly generated data allowed for more in-depth analysis. 

Processing of the sequencing data and contamination estimations are described well and met the established standards. The results attest that contamination is minimal. The authors employed two new approaches to estimate the relatedness between the two individuals (KIN and NgsRelate). These results support the previous findings based on READ and lcMLkin that individuals are second-degree relatives. As the relatedness was assessed by four different software it seems solid but I'm wondering how much these methods are independent one from another? Is the newly introduced ancIBD approach (10.1038/s41588-023-01582-w) different and is it possible to apply it to this case?

Most importantly newly generated data allowed to refine the Y chromosome haplogroup assignment and showed that individuals had distinct Y chromosomes so were not directly related in male line. Suggesting that Peder Winstrup was related to the foetus via his daughter.  What is also supported by elevated (or non-null) kinship coefficient on X chrom.

The conclusions of the study are supported by the results of the conducted analyses, and both sections are well and clearly presented.

I have a few minor suggestions:

line  61 - Wet laboratory processing - it is not entirely clear for me whether the description given applies to experiments performed for this or for the previous study. Later on, it becomes clear that previously produced libraries were sequenced, but maybe it is worth highlighting this here. 

line 171 - The integrity of the sequencing results is based on mtDNA data but the hq classification is provided only for merged data. Maybe it would be beneficial to also provide hg estimation based on newly generated data in Tab SX3A?  

line 188 - The first sentence about pathPhynder results is a repetition of the one in Materials and Methods.

Supplementary Table SX1 - column 'sequences' is it a number of newly generated sequences? Human - is this the number of human aligned reads merged from all experiments? Please provide more descriptive captions. 

line 248, Supplementary Table SX3 - for readers not being specialists in human Y chromosomes it is difficult to assess whether FTDNA and pathPhynder approaches resulted in the same haplogroup I suggest adding SNP based notation to the SX3C table if possible.

Author Response

Thank you for taking the time to review our article, for your positive assessment of our work and for your valuable comments.

We agree with the reviewer that using the newly introduced ancIBD approach (10.1038/s41588-023-01582-w) is different to the kinship estimation tools presented here and could constitute a useful addition to our study. That being said, we regretfully inform we are currently in the process of introducing ancIBD to our standard toolkit repertoire and are still testing its performance on our own datasets meaning we would not be able to add the analyses at this time.

Regarding the minor comments:

line  61 - Wet laboratory processing - it is not entirely clear for me whether the description given applies to experiments performed for this or for the previous study. Later on, it becomes clear that previously produced libraries were sequenced, but maybe it is worth highlighting this here

Thank you. We agree this should be made clearer and therefore we change the first sentence of “Wet laboratory processing” from “Sample handling and processing was described in detail in [1]. In short, the steps adopted included anthropological analysis based on a CT-scan produced at the Lund University Hospital [5]” to “The sample handling and processing procedures, outlined in detail in [1], were conducted as part of the earlier published study. In brief, the steps involved anthropological analysis utilizing a CT-scan generated at Lund University Hospital [5]”.

line 171 - The integrity of the sequencing results is based on mtDNA data but the hq classification is provided only for merged data. Maybe it would be beneficial to also provide hg estimation based on newly generated data in Tab SX3A?  

Thank you for this suggestion. For each of the two individuals we have now added subsection “win00x - newly generated data” where we add information regarding mtDNA assignment based on the new sequencing data in table SX3A.

line 188 - The first sentence about pathPhynder results is a repetition of the one in Materials and Methods.

Thank your for noticing. We now shorten the pathPhynder result section to avoid repetition. The paragraph now reads: “The pathPhynder (2020-12-19-b8532c0) best path method results suggest win001 could belong to lineage R1b1a1b1a1a2 or R1b1a1b1a1a2a1a1a1~, while win002 belonged to R1b1a1b1a1a2a1a2~ pointing to the possibility of the lineages being different (Table 1 and SX3). “

Supplementary Table SX1 - column 'sequences' is it a number of newly generated sequences? Human - is this the number of human aligned reads merged from all experiments? Please provide more descriptive captions. 

Thank you for noticing and mentioning this problem. We note the more descriptive captions are invisible due to inappropriate raw hight settings. We apologize and fix the issue.

line 248, Supplementary Table SX3 - for readers not being specialists in human Y chromosomes it is difficult to assess whether FTDNA and pathPhynder approaches resulted in the same haplogroup I suggest adding SNP based notation to the SX3C table if possible.

Thank you very much for this comment. Although it can be quite problematic to compare the two measures, we agree we should present the methods in as comparable and comprehensive way as possible. For the reason, in table SX3C, we have added a list of specific SNPs supporting the final haplogroup assignment as well as added the full list of all SNPs along their best paths and their respective counts identified in both individuals (untrimmed data).

Reviewer 2 Report

Comments and Suggestions for Authors

The article is clearly written and is supported by adequate approaches. It is a good learning. 

Author Response

Thank you very much for taking the time to read our manuscript and for your kind review. 

Reviewer 3 Report

Comments and Suggestions for Authors

Dear Editor of Heritage and Authors,

I have read the manuscript titled "Related in death? Further insights on the curious case of bishop Peder Winstrup and his grandchild’s burial" and, even though it features a follow-up from a previous investigation, I have read it with utmost interest and found it both scientifically compelling and accurate. I would only like to ask if there are any historical documentation regarding Anna Maria's pregnancy history and progeny that can illuminate the biography of the fetus found next to the bishop of Lund.

My best regards,

Author Response

Thank you for this comment. We agree with the Reviewer it would be useful if such documentation existed, however we are not aware of such documents.